# Intraoperative Blood Loss at Different Surgical-Procedure Stages during Posterior Spinal Fusion for Idiopathic Scoliosis

**DOI:** 10.3390/medicina59020387

**Published:** 2023-02-16

**Authors:** Eiki Shirasawa, Wataru Saito, Masayuki Miyagi, Takayuki Imura, Toshiyuki Nakazawa, Yusuke Mimura, Yuji Yokozeki, Akiyoshi Kuroda, Ayumu Kawakubo, Kentaro Uchida, Tsutomu Akazawa, Masashi Takaso, Gen Inoue

**Affiliations:** 1Department of Orthopaedic Surgery, School of Medicine, Kitasato University, Kanagawa 252-0374, Japan; 2Department of Orthopaedic Surgery, Medical Center, Kitasato University, Saitama 364-8501, Japan; 3Department of Orthopaedic Surgery, School of Medicine, St. Marianna University, Kanagawa 216-8511, Japan

**Keywords:** idiopathic scoliosis, posterior spinal fusion, blood loss

## Abstract

*Background and Objectives*: Several predictive factors have been reportedly associated with intraoperative total blood loss (TBL) during posterior spinal fusion (PSF) for idiopathic scoliosis (IS). To reduce TBL, preoperative factors and interoperative factors are considered important. However, there are few reports that have evaluated bleeding patterns according to surgical stages. This study aimed to elucidate bleeding patterns at different surgical stages and determine the predictive factors for TBL during PSF surgery in patients with IS. *Materials and Methods*: Preoperative data, radiographic parameters, and intraoperative data of patients undergoing PSF for IS were retrospectively collected. We divided the patients into six stages: stage 1, exposure; stage 2, implant placement; stage 3, release; stage 4, correction; stage 5, bone grafting; and stage 6, closure; then we reviewed the blood loss and bleeding speed. Multiple-regression analysis was performed to generate a predictive formula for blood loss using preoperative and intraoperative factors, including blood loss at stage 1, as explanatory variables. *Results*: Forty-five patients (mean age: 17.6 years) were included. The mean operative time and TBL were 287.9 min and 756.5 mL, respectively. Blood loss was the highest at stage 3, followed by stage 4. Bleeding speed was the highest at stage 4, followed by stage 3. Bleeding speeds at stages 3 and 4 were significantly higher than those at stages 1 and 2. Preoperative Cobb angle, activated partial thromboplastin time (aPTT), number of fused vertebrae, and blood loss at stage 1 were significant contributing factors. *Conclusions*: Blood loss and bleeding speed during the release and correction stages were high. Specifically, bleeding speed significantly increased during and after the release procedure. The preoperative Cobb angle, aPTT, number of fixed vertebrae, and blood-loss volume during PSF were significantly associated with TBL. Our findings would be helpful for reducing TBL in patients undergoing PSF for IS.

## 1. Introduction

Posterior spinal fusion (PSF) for idiopathic scoliosis (IS) can increase the risk of massive intraoperative blood loss, due to large skin incisions, bleeding from posterior elements such as bones and muscles, long surgical times, and general anesthesia. Massive blood loss could lead to perioperative cardiovascular and metabolic problems as well as coagulation disorders [1,2,3]. Regarding massive blood loss, allogeneic blood transfusion may be necessary, which could increase the risk of blood-borne viral infections, allergic reactions, and transfusion-associated graft-versus-host disease [4,5,6]. Additionally, patients with increased intraoperative blood loss show an increased prevalence of non-neurological postoperative complications such as respiratory problems, wound infections, and hematomas [2].

Both autologous and allogeneic blood transfusions are associated with increased morbidity and cost of hospital stays [5]. Therefore, the identification of perioperative risk factors for increased intraoperative blood loss and the ability to predict intraoperative total blood loss (TBL) are important for avoiding excessive blood loss and the need for allogeneic blood transfusion.

There have been many published studies on the perioperative risk factors associated with intraoperative blood loss and predictive factors and formulas for blood loss [7,8,9,10,11]. However, few reports have evaluated intraoperative blood loss during each surgical procedure, including opening, screwing, and release. Therefore, understanding the bleeding pattern at different stages would be important for reducing intraoperative TBL. Furthermore, the estimation of blood loss preoperatively or in the early phase of surgery would be helpful for reducing TBL and for increased surgical safety. The current study aimed to elucidate the bleeding patterns at different stages of surgery and determine the predictive factors of TBL during PSF surgery in patients with IS.

## 2. Materials and Methods

### 2.1. Patient Selection and Data Collection

The Institutional Review Board of our hospital approved this retrospective review of medical records. The study adhered to the tenets of the Declaration of Helsinki. This study included patients who underwent PSF for IS at our institution between March 2017 and March 2020. The inclusion criteria were (i) patients with a diagnosis of IS, (ii) age of 10–35 years at the time of surgery, and (iii) surgical correction that predominantly involved the use of a pedicle screw with or without hooks and/or sublaminar taping. The exclusion criteria were (i) any comorbidity affecting intraoperative bleeding, (ii) undergoing different surgical correction (mainly requiring hooks or tapes), and (iii) lack of important preoperative and intraoperative data. Figure 1 shows a flowchart of patient selection. Patient background and perioperative data collected retrospectively from the medical records of the patients included sex, age, height, weight, body mass index (BMI), operative time, and TBL.

The results of preoperative blood tests were also collected. They included hemoglobin (Hb), platelet count (Plt), prothrombin time (PT), activated partial thromboplastin time (aPTT), and fibrinogen (Fib) levels. A radiograph of the whole spine was used to obtain the Lenke classification, preoperative Cobb angle of the major curve, flexibility of the major curve, thoracic kyphosis (TK, measured between T5 and 12), number of fused vertebral bodies, number of pedicle screws, and correction rate. In addition, to understand the bleeding pattern at different PSF stages, we divided the patients into six stages. In stage 1 (exposure), incision of the skin and dissection of subcutaneous soft tissue and paravertebral muscles were performed to expose the posterior elements. During stage 2 (implant placement), pedicle screws were placed using a navigation-system assistance and hooks and sublaminar tapes were used, if needed. The inferior facet was resected routinely and superior-facet resection was performed at three or four intervertebral levels around the apex of the curve, depending on flexibility (stage 3, release). The correction procedure was performed using the single-rod rotation technique (stage 4, correction). After laminar and facet decortication, bone grafting was performed using a mixture of local autologous and artificial bone (stage 5, bone grafting). In stage 6 (closure), the drainage tube was placed in position, and the surgical wound was closed. Figure 2 shows the aforementioned six stages of PSF.

We also collected data on the procedure time and blood loss at each stage. As several authors have reported [12,13], we estimated blood loss on the cell saver and gauze at the end of each stage. The blood loss on the gauze was estimated as the difference between the weights of the dry and blood-soaked gauze. In addition, the blood loss on the cell saver was calculated according to the following formula.
The blood loss = final volume accumulated in the reservoir minus total volume of anticoagulant citrate dextrose minus the total irrigation fluid used intraoperatively.

Nurses recorded these volumes precisely on medical records, and two surgeons calculated and estimated blood loss at each stage. Then, the bleeding speed was calculated, which was divided by the procedure time and bleeding speed per fused level.

### 2.2. Surgical Procedure

A midline incision was made from the upper to lower vertebral levels, widely exposing the posterior elements of the spine. Whenever possible, pedicle screws were placed under the assistance of a navigation system (StealthStation S7 Surgical Navigation System [Medtronic, Inc., Minneapolis, MN, USA]). Additionally, data were acquired from a preoperative computed-tomography scan. Spinal correction was performed by inserting and rotating rods using appropriate distraction- and compression-based techniques. Hooks and sublaminar tape were used to correct spinal deformity and maintain correction. Autogenous local-bone grafts were harvested from the spinous processes, inferior facets, and transverse processes, according to need. Spinal-cord function was monitored using motor-evoked potentials throughout the procedure. Tranexamic acid was administered to all patients, according to the standard protocol of the Department of Anesthesia at our institution. All patients received a tranexamic-acid loading dose (20 mg/kg, 2 g maximum) immediately before skin incision. A maintenance drip (10 mg/kg/h; maximum of 5 g including the loading dose) was used throughout surgery. Intraoperative autotransfusion was performed in patients who were able to undergo preoperative blood donation. We also used a Continuous AutoTransfusion System (C.A.T.S.; Fresenius SE & Co., Bad Homburg vor der Höhe, Germany) for intraoperative blood collection and return.

### 2.3. Statistical Analysis

A repeated-measures analysis of variance (ANOVA) was performed, to compare the mean procedure time, blood loss, bleeding speed, and bleeding speed per fused level at each stage among the six groups (stages 1–6). Post hoc analysis was performed using the Bonferroni test for multiple comparisons. Regarding the estimation of blood loss preoperatively or in the early phase of surgery, stepwise-multiple-regression analysis was used to identify preoperative risk factors associated with TBL and calculate a predictive formula for TBL, with preoperative and intraoperative information designated as explanatory variables. Patient age, BMI, blood-test results (Plt count and Hb; PT; aPTT; and Fib levels), Cobb angle, and major-curve flexibility were used as preoperative background data to predict TBL. The number of fused vertebrae and pedicle screws and amount of blood loss during stage 1 were considered representative of surgical and intraoperative factors that could predict TBL. Statistical significance was set at *p* < 0.05. The Statistical Package for Social Sciences (SPSS) software (version 19.0; IBM Japan Business Services Co., Tokyo, Japan) was used for statistical analyses.

## 3. Results

A total of 45 (7 male, 38 female) patients were included in this study (mean age, 17.6 ± 5.0 years; height, 1.6 ± 0.1 m; and weight, 47.7 ± 7.6 kg). The mean preoperative Cobb angle and flexibility were 55.3 ± 11.5° and 43.9 ± 20.4%, respectively. Table 1 summarizes the demographic data and Lenke classifications.

The mean TBL volume was 756.5 ± 504.7 mL. The mean number of fused vertebrae was 9.7 ± 3.1. The mean postoperative Cobb angle and correction rate were 15.9 ± 8.6° and 72.2 ± 12.1%, respectively. The perioperative data are summarized in Table 2.

Considering the multiple comparisons of bleeding patterns at each stage, the mean procedure times at stages 1, 2, 3, and 4 were significantly higher than those at stages 5 and 6. In addition, the mean procedure times at stages 1 and 2 were significantly higher than that at stage 3. The mean procedure time at stage 5 was significantly shorter than that at stage 6 (Figure 3A). Regarding the blood loss at each stage, blood loss was the highest at stage 3, followed by that at stage 4. Blood loss at stages 1, 2, 3, and 4 was significantly higher than that at stage 6. Additionally, blood loss at stages 3 and 4 was significantly higher than that at stage 5 (Figure 3B). On the other hand, bleeding speed was the highest at stage 4, followed by that at stage 3. The mean bleeding speeds at stages 3 and 4 were significantly higher than those at stages 1, 2, and 6. In addition, there were significantly higher bleeding speeds at stage 5, compared with those at stages 2 and 6 (Figure 3C). Moreover, the bleeding speed per fused level was the highest at stage 4, followed by that at stage 3. The bleeding speed per fused level at stage 3 was significantly higher than that at stages 1, 2, and 6. In addition, the bleeding speeds per fused level at stages 4 and 5 were significantly higher than that at stage 6 (Figure 3D).

Multiple-regression analysis identified that preoperative Cobb angle, aPTT, number of fused vertebrae, and blood-loss volume at stage 1 were significant contributing factors to TBL. We created two predictive formulas for TBL, which were calculated as follows: TBL (mL) = −1167.3 + 15.2 × aPTT + 13.4 × preoperative Cobb angle (°) + 70.9 × number of fixed vertebrae (including preoperative factors), and TBL (mL) = −968.3 + 12.2 × aPTT + 10.8 × preoperative Cobb angle (°) + 54.2 × number of fixed vertebrae + 1.6 × blood loss during exposure (mL) (including preoperative factors and blood loss during the exposure stage) (Table 3).

## 4. Discussion

After analyzing the data of 45 patients with IS who underwent PSF, we found that the bleeding speeds at the stage of release and correction were significantly higher than those at the early stage of surgery, including exposure and implant placement. In addition, the preoperative Cobb angle and number of fused vertebrae were significantly associated with TBL. To estimate and reduce TBL, these preoperative factors, in addition to the amount of blood loss during the early stage of surgery, were used to develop two simple predictive formulas for TBL.

Regarding bleeding patterns at different stages of PSF surgery, two previous reports evaluated bleeding patterns at stage 1 and 2. Chiu et al. reported that in their study of patients with IS who underwent PSF surgery, the largest amount of blood loss was found at the stage of implant placement [14]. However, the bleeding speed was the highest during the stage of correction, followed by that at the stage of release, at stage 1. In contrast, Modi et al. studied adolescent-idiopathic-scoliosis (AIS) and neuromuscular-scoliosis cases, and found that blood loss was the highest during the stage of correction [15]. However, they did not evaluate bleeding speed at each stage. In the current study, we found that blood loss and bleeding rate were high during the release and correction stages, similar to that in previous reports. Specifically, bleeding speed was increased during and after the stage of release. These findings indicate that the procedure of release would be important for blood loss and bleeding speed during PSF surgery. Therefore, shortening the time at the stage of release and reducing the blood loss at the stage of release would be important for reducing TBL. Regarding the preventive and counteractive measures for preventing blood loss, we considered that reducing the blood loss at stages that indicated maximum blood loss and bleeding speed would be important. In the current study, the release and correction stage exhibited the maximum blood loss and bleeding speed. We used a gelatin-thrombin hemostatic matrix to reduce the blood loss from epidural veins during facet resection at the release stage. In their randomized clinical trial, Helenius et al. reported that a gelatin-thrombin hemostatic matrix decreased intra-operative blood loss by 30% in IS surgery [16]. Therefore, using appropriate hemostatic agents at appropriate stages would be important for reducing the amount of blood loss. Additionally, shortening the procedural time of release and correction is helpful in reducing the total amount of blood loss, because bleeding speed at the release and correction stage appears to be larger than those at the other stages. Therefore, we should plan a release strategy such that maximum release and correction effect with minimum procedural time would be achieved.

Published reports on preoperative factors have included patient sex, age, low BMI, aPTT, Cobb angle, and TK in the predictive formula for TBL [7,9,10,11,17]. Yu et al. evaluated 159 patients and found that a preoperative Cobb angle of >50° was a risk factor for massive TBL [11]. Nugent et al. reported that larger preoperative Cobb angles (particularly Cobb angles > 70°) in patients with AIS were associated with increased operative time and TBL [18]. Our study found a significant association between the preoperative Cobb angle and TBL, which is consistent with study reports in the literature [11,18]. In patients with larger preoperative Cobb angles, three-dimensional spinal deformities had progressed, which led to difficulties in performing the procedures, including dissection, release, and correction. Li et al. reported that hemostasis tended to be impaired during the premenstrual phase in patients and that aPTT values also increased. However, female patients with AIS tended to have an increased amount of TBL when surgery was performed during the premenstrual phase of their menstrual cycle [9]. Our results were consistent with those reported by Li et al., and the aPTT value was found to be one of the preoperative predictors. Alamanda et al. reported that male patients had a TBL that was 30% more than that in female individuals after adjusting for surgical time and BMI [17]. We did not observe a relationship between the sex of patients and TBL. Since there were only seven male patients in this study out of a total of forty-five patients, it might be possible that we may not have been able to determine whether the patient’s sex was a contributing predictive factor for TBL.

Meert et al. reported that patients with lower body weight may be at an increased risk of massive blood loss and possible need for blood transfusion during surgery for scoliosis [19]. They considered that surgery in patients with lower body weight was technically more difficult than in patients who were not underweight, due to increased bleeding or malnutrition in underweight patients. Malnutrition can affect the clotting function and exacerbate blood loss. In the present study, we did not observe a significant association between BMI and TBL. This may be explained by the relatively small sample size and minor variations in age and body weight of the patients in this study.

Oussama et al. reported that male patients with severe thoracic lordosis (T5–T12 ≤ 0°) were at risk of excessive intraoperative bleeding [20]. Deep dissection to expose the lordotic curve could have caused increased bleeding associated with TK. In our study, the association between extensive TK and blood loss was not statistically significant. Although some patients had thoracic lordosis (mean angle, 21°; range, 2–47°), none of our study patients had severe lordosis. The difference between the characteristics of the participating patients in our study and those in the study by Oussama et al. may account for the discrepancy between the results.

Some reports have identified intraoperative and surgical factors associated with increased TBL during spinal correction for AIS [7,10,18,19,21]. Yoshihara and Yoneoka reported that patients with AIS who underwent surgical procedures using a posterior approach and fusion of ≥9 levels were more likely to incur greater TBL and require allogeneic blood transfusions than those with fusion of 4–8 levels [21]. The results of this study are consistent with those of previous reports. We indicated that the number of fused vertebrae was a predictor of TBL. Ialenti et al. reported that the number of implanted anchors, such as pedicle screws, hooks, and wires, was an intraoperative variable that highly correlated with TBL [8]. Goobie et al. reported that tranexamic acid use in patients with AIS undergoing surgery significantly reduced blood loss, by 27%, compared with blood loss in the placebo group [22]. In this study, tranexamic acid was administered to all patients according to a standard protocol, and PSF was performed using standardized surgical procedures.

Two previous studies presented formulas that predicted TBL during PSF for AIS. Ialenti et al. found that preoperative TK, sex, and increased operative time were predictors of increased blood loss in patients with PSF. The following formula was calculated to estimate TBL: TBL (mL) = C + operative time (min) × (6.4) − preoperative T2 − T12 kyphosis (°) × (8.7), where C = 233 in male and C = −270 in female patients [8]. However, one limitation of this formula was that we were unable to predict TBL before the end of surgery because operative time was an independent variable that could not be obtained until after the surgical procedure. Chao et al. performed PSF in 161 female patients with AIS using an all-pedicle screw instrument. They attempted to predict the volume of surgical bleeding from the preoperative data of each patient [9]. Factors that strongly affect TBL included large fulcrum-bending Cobb angles, number of fused vertebrae, elevated Risser sign, elevated aPTT level, elevated preoperative blood-fibrinogen level, and phase of the menstrual cycle. All these factors were used to create a formula to predict TBL. Although the predictive factors included in this formula are similar to ours, their results might be limited by the fact that they only studied female patients, and analyzed the preoperative factors to identify predictors. Additionally, although we understand that the Risser sign could be an important predictor, obtaining an accurate assessment of the Risser sign was difficult in some patients [23]. In this study, we evaluated pre- and intraoperative factors associated with TBL in female and male patients, and we developed two predictive formulas: TBL (mL) = −1167.3 + 15.2 × aPTT + 13.4 × preoperative Cobb angle (°) + 70.9 × number of fixed vertebrae (including only preoperative factors), and TBL (mL) = −968.3 + 12.2 × aPTT + 10.8 × preoperative Cobb angle (°) + 54.2 × number of fixed vertebrae + 1.6 × blood loss during exposure (mL) (including preoperative factors and blood loss during the exposure stage). The predictive formula using only preoperative factors might be useful for estimating TBL preoperatively and determining the appropriate amount of stored preoperative autologous-blood donation. The predictive formula, including blood loss during the exposure stage, could be useful for estimating TBL during surgery and determining the need for allogeneic blood transfusion. Additionally, the accuracy of the predictive formulas was reflected by the adjusted R^2^ value, which improved from 0.50 to 0.59 after including intraoperative blood loss during the exposure stage. Compared with previous reports, and regardless of the sex or physique of the patients, our formulas were able to predict TBL simply and accurately when preoperative and surgical data from the early stage of the surgery were factored into our formula. We might contend that prediction of TBL using our formulas and sharing that information with the anesthesiologist and nurses in the operating room could improve the management of intraoperative blood loss and avoid excessive bleeding or the need for blood transfusion.

The current study has some limitations. First, the surgical strategies of PSF for IS vary among institutions and surgeons. In the current study, five surgeons were included. At the beginning of this case series, we tried to unify the surgical strategy relating to the order and manner of the procedure as much as possible. However, we were aware that the experience of the surgeons was not unified completely. Therefore, the results of this study, in particular our predictive formulas for TBL, are not completely applicable to every PSF for IS. In addition, there might be variation among institutions and surgeons. However, understanding the trend of bleeding at each stage, and the associated factors of TBL would be helpful for many surgeons in their efforts to reduce TBL. Additionally, findings that the volume of blood loss during the early stage of PSF was associated with TBL, in addition to the importance of calculating blood loss at each stage, may contribute to the improvement of the management of surgery for the medical staff in the operating room. Second, our study was a retrospective single-center study that evaluated a relatively small sample, and this was a critical issue of the current study. However, we suggest that the results of our study provide meaningful information because the predictive factors and formulas used for calculations in this study were simple and useful for clinical practice, particularly compared with formulas from previous reports [8,9]. Additional prospective studies are required to verify the validity of this formula.

## 5. Conclusions

Bleeding speed significantly increased during and after the procedure of release in patients undergoing PSF for IS. In addition, we found that the preoperative Cobb angle, aPTT, number of fixed vertebrae, and volume of blood loss during the exposure stage of PSF were significantly associated with TBL. Using our formula, TBL might be estimated preoperatively or in the early phase of surgery. These findings in the current study would be helpful in reducing TBL in patients undergoing PSF for IS.

## Figures and Tables

**Figure 1 medicina-59-00387-f001:**
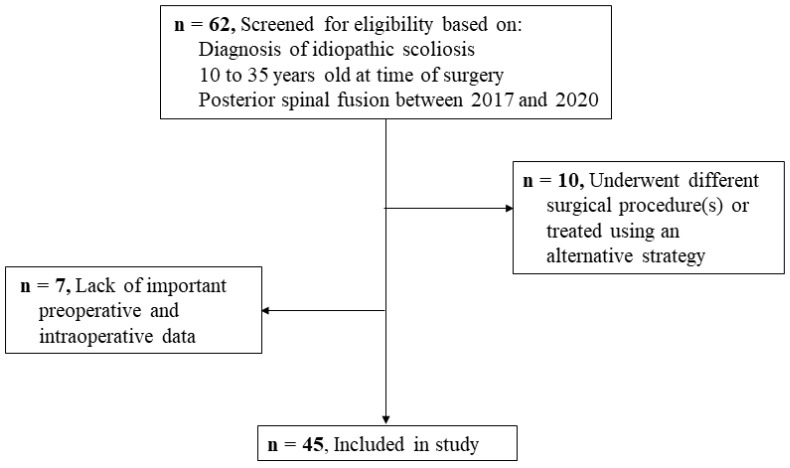
The flowchart of patient selection in this study. A total of 45 patients were finally included in this study.

**Figure 2 medicina-59-00387-f002:**
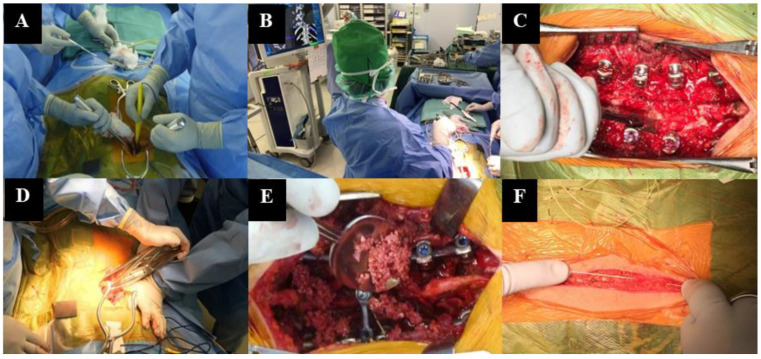
Six stages of PSF. (**A**) Exposure (stage 1), (**B**) Implant placement (stage 2), (**C**) Release (stage 3), (**D**) Correction (stage 4), (**E**) Bone grafting (stage 5), (**F**) Closure (stage 6).

**Figure 3 medicina-59-00387-f003:**
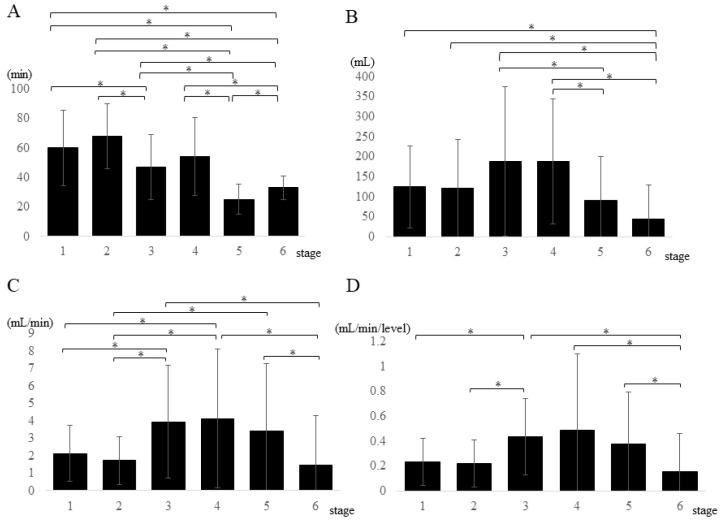
Multiple-comparison analysis of (**A**) procedure time, (**B**) blood loss, (**C**) bleeding speed, and (**D**) bleeding speed per fused level among the six stages are shown. * *p* < 0.05.

**Table 1 medicina-59-00387-t001:** Preoperative Demographic Data and Lenke Classification.

		*n* = 45
Sex		Male (*n* = 7); Female (*n* = 38)
Age (SD), years		17.6 (5.0)
Height (SD), m		1.6 (0.1)
Weight (SD), kg		47.7 (7.6)
BMI (SD), kg/m^2^		19.0 (2.5)
Hb (SD), g/dL		13.7 (1.6)
Plt (SD), ×10^4^/μL		26.2 (5.2)
PT (SD), s		12.6 (0.9)
aPTT (SD), s		32.7 (8.9)
Fib (SD), mg/dL		251 (44.4)
Preoperative Cobb angle (SD), °		55.3 (11.5)
Preoperative thoracic kyphosis (SD), °		21.1 (10.4)
Lenke type, number of patients	1	22
	2	5
	3	2
	4	1
	5	11
	6	4
Main Curve Flexibility (SD), %		43.9 (20.4)

Abbreviations: BMI, body mass index; Hb, hemoglobin; Plt, platelet count; PT, prothrombin time; aPTT, activated partial thromboplastin time; Fib, fibrinogen; SD, standard deviation.

**Table 2 medicina-59-00387-t002:** Summary of Perioperative Data.

Number of fused vertebrae	9.7 (3.1)
Postoperative Cobb angle (SD), °	15.9 (8.6)
Correction rate (SD), %	72.2 (12.1)
Operative time (SD), min	287.9 (68.2)
Total blood loss (SD), mL	756.5 (504.7)

SD, standard deviation.

**Table 3 medicina-59-00387-t003:** Multiple-Linear-Regression Analysis for Total Blood Loss. (**A**) Including only preoperative data. (**B**) Including preoperative data and blood loss during the exposure stage.

	Unstandardized		Standardized	t	Significance *p*
	B	SE	β		
Intercept	−1167.3	314.2			
aPTT	15.2	6.0	0.3	2.5	0.02 *
Preoperative Cobb angle	13.4	5.6	0.3	2.4	0.02 *
Fused vertebrae	70.9	21.2	0.4	3.3	<0.01 *
	Unstandardized		Standardized	t	Significance *p*
	B	SE	β		
Intercept	−968.3	293.5			
A aPTT	12.2	5.6	0.2	2.2	0.04 *
Preoperative Cobb angle	10.8	5.2	0.2	2.1	0.04 *
Fused vertebrae	54.2	20.1	0.3	2.7	0.01 *
Blood loss during exposure stage	1.6	0.5	0.3	3.1	<0.01 *

Abbreviations: aPTT, activated partial thromboplastin time; SE, standard error. * *p* < 0.05, R = 0.73, R^2^ = 0.50.

## Data Availability

The data presented in this study are available on request from the corresponding author.

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
