# Peer review of "Intraoperative Blood Loss at Different Surgical-Procedure Stages during Posterior Spinal Fusion for Idiopathic Scoliosis"

_medicina, 2023, doi:10.3390/medicina59020387_

Round 1

Reviewer 1 Report

The authors of the study evaluated bleeding rate and total blood loss at different stages of AIS correction. The authors retrospectively evaluated 45 patients who underwent surgery during a 3-year time period.

I have the following concerns with this manuscript.

It is difficult to understand how the authors accurately estimated blood loss at each stage. Was it estimated through the suction or through cell saver? Was it purely an estimate? Were all the estimates performed by the same author or were different surgeons estimating the blood loss? 

It is difficult to understand the clinical relevance of when the authors experienced the most blood loss and if this is consistent across different surgeons or even surgeons within the same author group if multiple surgeons had data collected. Some surgeons may be less likely to achieve hemostasis during exposure and lose more blood, while others may lose more during facetectomy. Some may be more likely to use hemostatic agents like surgiflow or bone wax, which may reduce blood loss. All these factors make it difficult to interpret the authors data. This would probably require prospective analysis. 

Author Response

Response to reviewer’s comment

We thank the reviewer for taking time to review our paper. As reviewer’s suggestion, we revised our manuscript as follows.

Reviewer #1

  1. It is difficult to understand how the authors accurately estimated blood loss at each stage. Was it estimated through the suction or through cell saver? Was it purely an estimate? Were all the estimates performed by the same author or were different surgeons estimating the blood loss? 

Response to comment from Reviewer #1-1

We thank the reviewer for this suggestion.

We usually measure and record the total amount of blood loss on the cell saver and gauze at the end of each stage in scoliosis surgery for taking advantage of stable anesthetic management. In the current study, we used these records. We believe that this measurement method was the most objective and accurate way of estimating the blood loss. We added these following sentences to the Materials and Methods.

We measured the total amount of blood loss on the cell saver and gauze at the end of each stage.

  1. It is difficult to understand the clinical relevance of when the authors experienced the most blood loss and if this is consistent across different surgeons or even surgeons within the same author group if multiple surgeons had data collected. Some surgeons may be less likely to achieve hemostasis during exposure and lose more blood, while others may lose more during facetectomy. Some may be more likely to use hemostatic agents like surgiflow or bone wax, which may reduce blood loss. All these factors make it difficult to interpret the authors data. This would probably require prospective analysis. 

Response to comment from Reviewer #1-2

We totally agree with the reviewer's suggestion, and we consider this point is one of the limitations of this study. In the current study, five surgeons were included. Before the beginning of this case series, we unified completely surgical strategy including order and manner of the procedure. We used hemostatic agents including gelatin-thrombin hemostatic matrix and bone wax in all cases. However, we could not unify the experience of surgeon. Therefore, we also considered this point was critical limitation of the current study. We added these following sentences to discussion section.

First, the surgical strategies of PSF for IS vary among institutions and surgeons. In the current study, five surgeons were included. At the begining this case series, we tried to unify the surgical stragedy including order and manner of the procedure as much as possible. However, we aware that the experience of surgeon was not unified completely. Therefore, the results of this study were not completely applicable to every PSF for IS. However, understanding the results of increased bleeding speed during and after the release procedure would be helpful for many surgeons to reduce the TBL.

Please see the attachment of main text.

Reviewer 2 Report

Dear Authors 

I commend your efforts to estimate the blood loss in relation to the stage of surgery in idiopathic scoliosis. I have some concerns regarding the methodology that prevent me from considering the paper for publication. 

Although the results of the study are more evident that the maximum blood loss would be in the stage of release and correction, the methodology of the study done retrospectively concerns me on the validity of the data obtained on blood loss flow rate during each stage of surgery. 

Also, the authors could also discuss on the preventive and counteractive measures for preventing blood loss in the stage proposed as a maximum loss.

Also, the authors could validate the formulas presented in their prospective cases to make them more practical rather than an outcome of statistical analysis. 

Author Response

Response to reviewer’s comment

We thank the reviewer for taking time to review our paper. As reviewer’s suggestion, we revised our manuscript as follows.

Reviewer #2

  1. Although the results of the study are more evident that the maximum blood loss would be in the stage of release and correction, the methodology of the study done retrospectively concerns me on the validity of the data obtained on blood loss flow rate during each stage of surgery. 

Response to comment from Reviewer #2-1

We thank the reviewer for this suggestion.

We usually measure and record the total amount of blood loss on the cell saver and gauze at the end of each stage in scoliosis surgery for taking advantage of stable anesthetic management. In the current study, we used these records. We believe that this measurement method was the most objective and accurate way of estimating the blood loss. We added these following sentences to the Materials and Methods.

We measured the total amount of blood loss on the cell saver and gauze at the end of each stage.

  1. Also, the authors could also discuss on the preventive and counteractive measures for preventing blood loss in the stage proposed as a maximum loss.

Response to comment from Reviewer #2-2

We thank the reviewer's suggestion. We also considered this is important point of this study. In the current study, release and correction stage showed the maximum blood loss and bleeding speed. We usually used gelatin-thrombin hemostatic matrix for reducing the blood loss from epidural venous during facet resection at release stage. Additionally, we considered shortened the procedure time of release and correction would be helpful for reducing total amount of blood loss, because bleeding speed at release and correction stage showed larger than those at the others stage. We added these following sentences and reference to discussion section.

Regarding the preventive and counteractive measures for preventing blood loss, we considered reducing the blood loss at stages showed maximum blood loss and bleeding speed would be important. In the current study release and correction stage showed the maximum blood loss and bleeding speed. We usually used gelatin-thrombin hemostatic matrix for reducing the blood loss from epidural venous during facet resection at release stage. Helenius et al reported in their randomized clinical trial, gelatin-thrombin hemostatic matrix decreases intra-operative blood loss by 30% in IS surgery. Therefore, using appropriate hemostatic agents at appropriate stage would be important for reducing the amount of blood loss. Additionally, shortened the procedure time of release and correction would be helpful for reducing total amount of blood loss, because bleeding speed at release and correction stage showed larger than those at the others stage. Therefore, we should plan the release strategy, so that maximum release and correction effect with minimum procedure would be achieved.

Helenius I, Keskinen H, Syvänen J, Lukkarinen H, Mattila M, Välipakka J, Pajulo O. Gelatine matrix with human thrombin decreases blood loss in adolescents undergoing posterior spinal fusion for idiopathic scoliosis: a multicentre, randomised clinical trial.

Bone Joint J. 2016 Mar;98-B(3):395-401.

  1. Also, the authors could validate the formulas presented in their prospective cases to make them more practical rather than an outcome of statistical analysis. 

Response to comment from Reviewer #2-3

We totally agree with reviewer's suggestion. We also considered this point was limitation of the current study and stated about this point in limitations section as follows. As reviewer's suggestion, we would like to validate these formulas in further prospective cases.

Second, our study was a retrospective single-center study that evaluated a relatively small sample, which was a critical issue of the current study. However, we suggest that the results of our study provide meaningful information because the predictive factors and formulas used for calculations in this study were simple and useful for clinical practice, particularly compared with formulas from previous reports [8, 9]). Additional prospective studies are required to verify the validity of this formula.

Please see the attachment of main text.

Round 2

Reviewer 1 Report

 The following concerns still exist: (1) A calculated blood loss is not utilized along with an estimated blood loss. (2) Surgeon variability may result in differences in bleeding speed and total blood loss at each stage. Even if unified within the surgeons groups, this may differ across hospital settings. 

Author Response

Response to reviewer

  1. A calculated blood loss is not utilized along with an estimated blood loss.

Response to comment from Reviewer #1-1

We thank the reviewer for this suggestion. We usually estimated the blood loss on the cell saver by the formula as previously reported.

As reviewer's suggestion, we added these following sentences to the Materials and Methods.

As several authors reported [12] [13], we estimated blood loss on the cell saver and gauze at the end of each stage. The blood loss on gauze was estimated the difference between the weights of dry and blood-soaked gauze. In addition, the blood loss on cell saver was calculated according to the formula as follows,

The blood loss = Final volume accumulated in the reservoir minus Total volume of anticoagulant citrate dextrose minus Total irrigation fluid used intraoperatively

Nurses recorded these volumes on medical records strictly, and two surgeons calculated and estimated blood loss at each stage.

[12] Duramaz A, Bilgili MG, Bayram B, Ziroglu N, Edipoglu E, Ones HN, et al. The role of intraoperative cell salvage system on blood management in major orthopedic surgeries: a cost-benefit analysis. Eur J Orthop Surg Traumatol. 2018;28:991-7.

[13] Nadler SB, Hidalgo JH, Bloch T. Prediction of blood volume in normal human adults. Surgery. 1962;51:224-32.

  1. Surgeon variability may result in differences in bleeding speed and total blood loss at each stage. Even if unified within the surgeons groups, this may differ across hospital settings.

Response to comment from Reviewer #1-2

We thank the reviewer for this suggestion. As mentioned in revise of round 1, we totally agree with the reviewer's suggestion, and we consider this point is one of the limitations of this study. We also understood the results of this study were not completely applicable to every PSF for IS. Our predictive formulas for total blood loss in our paper was just a reference in our institution. However, we considered the important points of findings and the significance of the current study were as follows,

  1. Bleeding speed significantly increased during and after the procedure of release in patients undergoing PSF for IS.
  2. The preoperative Cobb angle, aPTT, and number of fixed vertebrae were significantly associated with total blood loss.
  3. The volume of blood loss during the exposure stage of PSF might estimate total blood loss.

These findings would be helpful for many surgeons to reduce the TBL.

We added these following sentences to discussion section.

Therefore, the results of this study, in particular our predictive formulas for TBL are not completely applicable to every PSF for IS. Also, there might be variation among institutions and surgeons. However, understanding the trend of bleeding at each stage, and associated factors of TBL would be helpful for many surgeons in their efforts to reduce TBL. Additionally, findings that the volume of blood loss during the early stage of PSF was associated with TBL, in addition to the importance of calculating blood loss at each stage may contribute to the improvement of the management of surgery for medical staffs in operating room.

Please see the attachment for the revised text.

Reviewer 2 Report

I find that authors have addressed my queries raised in the previous rounds of review and hence I recommend the manuscript now for publication. 

Round 3

Reviewer 1 Report

NA